# CHyLL: Learning Continuous Neural Representations of Hybrid Systems

## Abstract

Learning the flows of hybrid systems that have both continuous and discrete time dynamics is challenging. The existing method learns the dynamics in each discrete mode, which suffers from the combination of mode switching and discontinuities in the flows. In this work, we propose **CHyLL** (**C**ontinuous **Hy**brid System **L**earning in **L**atent Space), which learns a continuous neural representation of a hybrid system without trajectory segmentation, event functions, or mode switching. The key insight of CHyLL is that the reset map glues the state space at the guard surface, reformulating the state space as a piecewise smooth quotient manifold where the flow becomes spatially continuous. Building upon these insights and the embedding theorems grounded in differential topology, CHyLL concurrently learns a singularity-free neural embedding in a higher-dimensional space and the continuous flow in it. We showcase that CHyLL can accurately predict the flow of hybrid systems with superior accuracy and identify the topological invariants of the hybrid systems. Finally, we apply CHyLL to the stochastic optimal control problem.

## 1 Introduction

Hybrid systems provide a powerful mathematical framework for modeling a broad spectrum of complex dynamics, where the evolution of states is governed by an interplay between continuous-time dynamics and discrete event-driven transitions. Such systems naturally arise in diverse applications, including rigid-body contact dynamics in robotics (Posa et al., 2014; Westervelt et al., 2003), large-scale traffic flow networks (Gomes & Horowitz, 2006; van den Berg et al., 2016), molecular interactions in biophysics (Anderson et al., 2007; Takada, 2015), and the coordination of humanoid motions (Ames et al., 2014; Grizzle et al., 2001). The hybrid formulation captures both continuous flows and abrupt state changes, enabling precise descriptions of systems that cannot be adequately represented by purely continuous or purely discrete models.

While the hybrid nature offers exceptional expressive power, it is challenging for controller design, verification, and learning the underlying dynamics from data. The primary difficulty stems from the intrinsic discontinuities induced by discrete state transitions—such as impacts, switches, or mode changes. These discontinuous breaks the smoothness assumptions that underpin many conventional learning algorithms for dynamical systems. On the other hand, the number of modes or possible transitions for each trajectory also results in an exponential number of combinations that are intractable for system identification when we learn the dynamics in the original state space.

In this work, we propose **CHyLL** (**C**ontinuous **Hy**brid System **L**earning in **L**atent Space) to learn the hybrid systems from only time series data. We show that exploiting the topological structure of hybrid systems enables one to learn the flow of hybrid systems via only continuous functions. The key insight stems from hybrid system theory (Simic et al., 2005) where the guard surface, i.e., the surface where the discrete changes happen, can be glued by the reset map to reformulate the entire state space as a piecewise smooth quotient manifold. On this quotient manifold, the flow of the hybrid systems becomes continuous, which is more suitable for a differentiable learning pipeline. While the topological theorem in (Simic et al., 2005) proved the existence of such a quotient manifold, there lacks a systematic way to construct it for numerical computations. To mitigate this gap, we further leverage the embedding theorem to learn a quotient manifold in a

Figure 1: CHyLL framework. The discontinuity of the flow of hybrid systems makes it hard to learn the dynamics. We introduce a continuous learning framework that reformulates the dynamics on a piecewise smooth manifold by gluing the surface where the mode change happens. We introduce a dual training strategy that learn the continuous flow in a higher-dimensional space without singularity and then decode it to the original state space.

singularity-free manner. The main structure of the proposed framework is illustrated in Figure 1. In summary, the main contributions of this work are:

1. Formulate the problem of learning the dynamics of a hybrid system from time series data as supervised learning on an unknown piecewise smooth manifold,

2. Propose the CHyLL framework that learns the continuous manifold representation and the dynamics of a hybrid system concurrently using only time-series data without trajectory segmentation, event functions, or mode switching.

3. Showcase CHyLL on learning hybrid systems, exploring the topological invariants, and applications in stochastic optimal control.

## 2 RELATED WORK

### 2.1 LEARNING HYBRID SYSTEMS

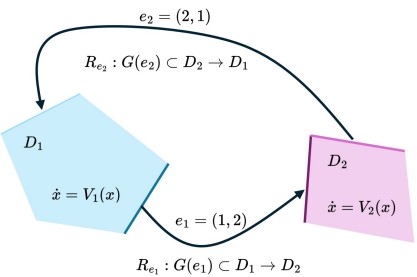

Figure 2: An example of a hybrid automaton. The state space of the hybrid system can be partitioned into different domains $D_i$ governed by a continuous vector field $V_i(\cdot)$. When the state reaches the guard $G(\cdot)$, the state will be mapped to other domains specified by the edges $e_i$ via the reset function $R(\cdot)$.

The Neural ODE by Chen et al. (2018) has been proposed to learn the continuous-time vector field from the time series observations of the flows generated by ordinary differential equations. However, when learning hybrid systems, Neural ODE fails to learn the discontinuous mode change as the vector field is represented by neural networks that can not uniformly approximate the discontinuous functions. To mitigate this issue, the event Neural ODE (Chen et al., 2020) is proposed to learn the hybrid automaton model by introducing additional event functions and the reset maps. Though these methods mimicked the structure of the hybrid automaton and thus inherently have the discontinuous structure, Chen et al. (2020) suffers from the sparsity of mode changes. If the mode change never happen or the initial solution is ill-posed, the event Neural ODE does not work. Similarly, the Neural Hybrid Automata is proposed in Poli et al. (2021) to learn stochastic hybrid systems represented by the dynamics module for the continuous dynamics, the discrete latent selector for the mode, and the event module for the transition between modes. Its discrete-time counterpart, the neural discrete hybrid automata, is proposed in Liu et al. (2025) to enable agile motor skills for legged robots with rich contact interactions. As a mixture-of-expert structure, both (Liu et al., 2025) and (Poli et al., 2021) requires a maximal number of neural networks for the dynamics in each mode. For systems with contact, the structure of the Linear Complementarity Problem (LCP) has been integrated into the learning pipeline. More recent work focuses on differentiating through the LCP (Bianchini et al., 2023; 2025; Jin et al., 2022; Pfrommer et al., 2021; Yang et al., 2025) to avoid the use of event functions. The LCP-based method can be considered as a geometric description of the system, thus naturally avoiding the combination of the modes.

Other than the hybrid automaton or geometric formulations like LCPs, the topological structure of hybrid systems has been widely studied in the control community. The *hybridfold* is proposed in (Simic et al., 2005) to convert the hybrid automaton to a single unified manifold by gluing the guard surface using the equivalence relationship defined by the reset map. A metrization method is later proposed in (Burden et al., 2015) to define the distance between trajectories on the glued space. Though these methods are topologically insightful, they do not show how to formulate the manifolds for numerical optimization. Nonetheless, these methods provide a promising direction to combine with learning and on-manifold optimization techniques.

## 2.2 TOPOLOGICAL & GEOMETRIC LEARNING

The manifold structure of data has been extensively studied in the machine learning community. Roweis & Saul (2000); Tenenbaum et al. (2000) embed the data into a single manifold in an unsupervised and non-parametric manner. In addition to a global embedding, *atlas learning* has been applied in (Pitelis et al., 2013) and (Cohn et al., 2022) to learn a piecewise embedding of the manifold that can potentially preserve the topological information. In addition to discovering the underlying manifold structure, the Lie group structure has been enforced in (Deng et al., 2021; Esteves et al., 2018; Lin et al., 2023) to enable higher data efficiency in 3D perception tasks. To learn the normalizing flow on a known manifold, Lou et al. (2020) extends the Neural ODE to non-Euclidean space via learning the vector field in a local chart.

Data-driven methods have been applied to discover the underlying topological structure of data. The *persistent homology* (Edelsbrunner & Harer, 2010) is proposed to discover the topological invariants from point clouds. This method detects multi-scale topological invariants by forming a simplicial complex. By the *persistence diagram*, we can extract the topological invariance of the data. The persistence image, a finite dimensional vector representation of the diagram, is proposed in Adams et al. (2017) for classification tasks. Zhou et al. (2019) explored the topology of real projective space to represent rotations in 3D perception without singularities. For learning of dynamics, the persistent homology is applied to time-series in (Perea & Harer, 2015) to explore the periodicity of the data. Moor et al. (2020) proposed the Topological Autoencoders to first explore the topological invariance in the input data and then add regularization to preserve the discovered connectivity information. To learn time-series data with intersections in the flow, the Augmented Neural ODE Dupont et al. (2019) appends additional latent dimensions to the vector field to lift the flow to a higher-dimensional space that does not have such intersections.

## 3 PRELIMINARIES

Consider a finite-dimensional smooth manifold $M$. The tangent space at a point $x \in M$ is denoted by $\mathrm{T}_x M$. The tangent bundle $\mathrm{T} M := \bigcup_{x \in M} \mathrm{T}_x M$ is the disjoint union of tangent spaces. A (smooth) vector field is a map $V : M \to \mathrm{T} M$ such that $V(x) \in \mathrm{T}_x M$ for all $x \in M$. The set of all smooth vector fields on $M$ is denoted by $\mathfrak{X}(M)$. A curve $c : (t_0, t_1) \to M$ is said to be the *integral curve* of the vector $V$ if $\dot{c}(t) = V(c(t))$. The integral curves by $V$ define the flow $\Phi(t, x) : \mathbb{R} \times M \to M$ that indicates the point $c(t)$ at time $t$ with initial condition $c(0) = x \in M$ and satisfy the group law $\Phi(t_2, \Phi(t_1, x_0)) = \Phi(t_1 + t_2, x_0)$.

The hybrid automaton is illustrated in Figure 2. Where we see that the flow can be discontinuous when the state hit certain submanifolds of the state space. We now refer to the (Simic et al., 2005) for standard mathematical definition of the hybrid systems.

**Definition 1** (hybrid systems (Simic et al., 2005)). *A hybrid system defined on $M$ is a 6-tuple*

$$\mathcal{H} = (Q, E, \mathcal{D}, \mathcal{V}, \mathcal{G}, \mathcal{R}), with$$

- $Q = \{1, \ldots, k\}$ *is the finite set of (discrete) states, where $k \geq 1$ is an integer;*

- $E \subset Q \times Q$ *is the collection of edges;*

- $\mathcal{D} = \{D_i : i \in Q\}$ *is the collection of domain, where $D_i \subset \{i\} \times M$, for all $i \in Q$;*

- $\mathcal{V} = \{V_i : i \in Q\}$ *is the collection of vector fields such that $V_i$ is Lipschitz on $D_i$, $\forall i \in Q$;*

- $\mathcal{G} = \{G(e) : e \in E\}$ *is the collection of guards, with $\forall e = (i, j) \in E, G(e) \subset D_i$;*

- $\mathcal{R} = \{R_e : e \in E\}$ *is the collection of resets, where* $\forall e = (i, j) \in E, R_e$ *is a relation between elements of* $G(e)$ *and elements of* $D_j$, *i.e.,* $R_e \subset G(e) \times D_j$.

We then define the time trajectories to indicate for the flow in each domain:

**Definition 2** (Hybrid time trajectory (Simic et al., 2005))**.** *A (forward) hybrid time trajectory is a sequence (finite or infinite)* $\tau = \{I_j\}_{j=0}^N$ *of intervals such that* $I_j = \left[\tau_j, \tau_j'\right]$ *for all* $j \geq 0$ *if the sequence is infinite; if* $N$ *is finite, then* $I_j = \left[\tau_j, \tau_j'\right]$ *for all* $0 \leq j \leq N - 1$ *and* $I_N$ *is either of the form* $[\tau_N, \tau_N']$ *or* $[\tau_N, \tau_N')$. *Furthermore,* $\tau_j \leq \tau_j' = \tau_{j+1}$, *for all* $j$.

The execution or the flow of $\mathcal{H}$ is defined as:

**Definition 3** (Flow of $\mathcal{H}$ (Simic et al., 2005))**.** *An execution of a hybrid system* $\mathcal{H}$ *is a triple* $\chi = (\tau, q, x)$, *where* $\tau$ *is a hybrid time trajectory,* $q : \langle \tau \rangle \to Q$ *is a map, and* $x = \{x_j : j \in \langle \tau \rangle\}$ *is a collection of* $C^1$ *maps such that* $x_j : I_j \to D_{q(j)}$ *and for all* $t \in I_j$, $\dot{x}_j(t) = V_{q(j)}(x_j(t))$. *The flow of the hybrid system* $x(t) = \Phi(t, x_0)$ *satisfy* $\dot{x}_j = V_{q(j)}(\Phi(t, x_0)), t \in I_j$.

Thus, we see that the trajectories of $\mathcal{H}$ in the original state space can be discontinuous at $\tau_k'$ and $\tau_{k+1}$ due to the reset functions. An example fo the hybrid system is shown in Figure 2. To avoid pathological behavior, we also require the systems to satisfy several regularity conditions, which we defer to Appendix A.

## 4 PROBLEM FORMULATION

We formally define the problem of learning hybrid systems from time-series data.

**Problem 1** (Learning hybrid system from time series data)**.** *Consider time-series observation of system* $\mathcal{H}$ *indicated by* $q$ *with length* $T$ *as* $\gamma_q = \{(t_0^q, x_0^q), (t_1^q, x_1^q), (t_2^q, x_2^q), \cdots, (t_T^q, x_T^q)\}$ *with each* $x_k^q \in M$ *recorded at time* $t_k^q$. *Denote a date set with* $N$ *trajectories as* $\mathcal{X} := \{\gamma_q\}_{q=1}^N$. *Our goal is to learn the flow of* $\mathcal{H}$ *from* $\mathcal{X}$.

The key challenge of learning the hybrid system $\mathcal{H}$ originates from the reset maps $R_e$ that instantaneously map the state at the guard surface $x^- \in G_e$ to its image via $x^+ = R_e(x^-)$. Such discrete jumps make the flow $\Phi(t, x)$ non-smooth or even discontinuous. From the conventional perspective of hybrid systems that evaluate the systems in each domain separately, the flow at the time of reset may not be a conventional function but an impulse distribution, which is hard to learn using continuous neural networks.

Though $\mathcal{H}$ contains the index set $Q$, guard surface $\mathcal{G}$, and the reset map $\mathcal{R}$, this information is usually more difficult to measure and thus this work does **not** assume any knowledge other than the time series data $\mathcal{X}$ observed on the flow $\Phi(\cdot, \cdot)$. We also note that recovering $\Phi(\cdot, \cdot)$ does not require an explicit representation of $\mathcal{G}$ or $\mathcal{R}$, which suffers from the combinatorially many mode selections, and also unnecessary, as shown in Simic et al. (2005) and this work.

## 5 CONTINUOUS HYBRID SYSTEM LEARNING IN LATENT SPACE

### 5.1 GLUING THE CONFIGURATION SPACE

To mitigate the discontinuity of $\Phi$, we construct the quotient manifolds induced by the reset map and learn $\Phi(\cdot, \cdot)$ on it. Now we apply the techniques from Simic et al. (2005) to generate the quotient manifold, namely *hybrifold*, i.e., a piecewise smooth manifold-like structure for $\mathcal{H}$:

**Definition 4** (Hybrifold (Simic et al., 2005))**.** *Let* $\mathcal{H}$ *be a hybrid system. On the* $n$ *dimensional manifold* $M$, *let* $\sim$ *be the equivalence relation generated by* $x \sim \tilde{R}_e(x)$, *for all* $e \in E$ *and* $x \in \overline{G(e)}$. *Collapse each equivalence class to a point to obtain the quotient space*

$$M_{\mathcal{H}} = M/ \sim . \tag{Hybrifold}$$

**Theorem 1** (Smoothness of Hybrifold (Simic et al., 2005))**.** $M_{\mathcal{H}}$ *is a topological* $n$*-manifold*[1] *with boundary, and both* $M_{\mathcal{H}}$ *and its boundary are piecewise smooth* [2].

---

[1] A topological $n$-manifold is a Hausdorff, second-countable topological space, which is continuous.

[2] Smooth in countably many partitioned regions and only non-smooth at the boundaries.

When the graph $(\mathcal{D}, E)$ is connected, $M_{\mathcal{H}}$ is connected where the flow is continuous. We note that Theorem 1 is only a topological argument. Even when $\mathcal{H}$ is fully known, a parameterized singularity-free continuous representation of the hybrifold $M_{\mathcal{H}}$ is unknown and may not be unique. To the best of the author's knowledge, there has been no systematic way to construct $M_{\mathcal{H}}$, either through analytical or data-driven methods.

## 5.2 Continuous Latent Space Embedding

In this work, we propose to concurrently learn $M_{\mathcal{H}}$ and $\Phi(\cdot, \cdot)$ using only the time series data $\mathcal{X}$. The first challenge is to formulate a global embedding of $M_{\mathcal{H}}$ without singularities. As shown in many classical examples in topology, the glued manifold with dimension $n$ does not have a global smooth representation in $\mathbb{R}^n$. For example, the torus $\mathbb{T}^2$ and the Klein bottle $\mathbb{K}$ can both be constructed by gluing the edges on $[0, 1]^2$, while $\mathbb{T}^2$ has to be in $\mathbb{R}^3$ and $\mathbb{K}$ to be in $\mathbb{R}^4$ to have singularity-free representations without self-intersections. In this work, instead of learning the dynamics on local charts (Lou et al., 2020) or on an atlas (Cohn et al., 2022), we learn a global representation of the hybrifold, which is guaranteed to exist and has a finite-dimensional embedding:

**Theorem 2** (Whitney Embedding Theorem (Hirsch, 2012)). *Any $C^r$-manifold[3] $M$ $(r \geq 1)$ of dimension $n$ can be embedded into $\mathbb{R}^{2n}$.*

**Remark 1** (Global Continuous Structure). *For $M_{\mathcal{H}}$ **without** a global $C^1$ structure, such as the case with corners on the boundaries, $M_{\mathcal{H}}$ is still continuous (global $C^0$ is always possible by Theorem 1) and can be embed into $\mathbb{R}^{2n+1}$ by Menger–Nöbeling theorem.*

The Whitney embedding theorem suggests that there exists a smooth injective function that maps $\forall x \in M$ to the ambient space $Z := \mathbb{R}^m$ with $m \geq 2n$:

$$E(\cdot) : M \to Z, \tag{1}$$

For $z \in Z$ not on the boundary, we can recover the unique $x \in M$ by the inverse:

$$E^{-1}(\cdot) : \text{Img } E \to M. \tag{2}$$

Given this insight, we propose to encode the data into a higher-dimensional latent space where the hybrifold is guaranteed to have a global continuous representation.

**Remark 2.** *Conventional methods compress the original data to a lower-dimensional embedding, while we note that it has no guarantee that the topological structure can be preserved. In our work, we show that increasing the dimension can help preserve the topological structure, which is the key to learn the discontinuous flow.*

**Remark 3.** *Though $E(\cdot)$ is a smooth function, $E^{-1}(\cdot)$ can be discontinuous. One example is the 1-D torus $\mathbb{T}^1 \simeq \text{SO}(2)$. The 1-D parameterization with $E(\cdot) : t \to (\sin t, \cos t)$ is smooth, while the inverse $E^{-1}(\cdot) : (\sin t, \cos t) \to \text{atan2}(\sin t, \cos t)$ has discontinuities.*

## 5.3 Main Algorithm

Given the analysis in the last section, we proceed to learn the embedding $E(\cdot)$, its inverse $E^{-1}(\cdot)$ and the flow $\Phi(\cdot, \cdot)$ in the latent space. We consider the encoder $E_\theta(\cdot)$ as the smooth embedding from the original state space $M$ to the latent space $Z$; The flow on $Z$ as $\Phi_\theta(t, z)$; and the inverse of $E_\theta(\cdot)$ is $D_\xi(\cdot)$: $z = E_\theta(x) : M \to Z, \quad x = D_\xi(z) : Z \to M, \quad \Phi_\theta(t, x) : \mathbb{R} \times Z \to Z.$

By Remark 3, we see that the flow and encoder are continuous functions, while the decoder can be discontinuous. As the discontinuity may result in unstable training behavior, we decouple the training of continuous and discontinuous components. Thus, we group the parameters for the flow and encoder as $\theta$, and denote the parameters for the decoder as $\xi$.

To train $E_\theta(\cdot)$ and $\Phi_\theta(\cdot, \cdot)$, we propose to optimize the continuous loss function, namely $\mathcal{L}_c(\theta)$:

$$\mathcal{L}_c(\theta) = w_1 \underbrace{\text{MSE}\big(E_\theta(x_k^q), z_k^q\big)}_{\text{Dynamics Loss over } (q, k)} + w_2 \underbrace{\text{MSE}\big(E_\theta(x_k^q), E_\theta(x_{k+1}^q)\big)}_{\text{Gluing Loss over } (q, k \in \mathcal{C})}$$

$$+ w_3 \underbrace{\text{MSE}(\theta_c I_n - (\frac{\partial E}{\partial x})^\top \frac{\partial E}{\partial x}|_{x_k^q})}_{\text{Conformal Loss}} + w_4 \underbrace{\sum_{i=1}^m \text{ReLu}\big(\Lambda - \text{Cov}(E_\theta(x)_i)\big)}_{\text{Latent Collapse Loss}}, \tag{3}$$

---

[3]Theorem 1 is a special case when $r = \infty$.

with the flow in the latent space parameterized by the vector field $V_\theta(\cdot) : Z \to Z$:

$$z_k^q = \Phi_\theta(t_k, E_\theta(x_0^q)) = \int_0^{t_k} V_\theta(z(t))dt + E_\theta(x_0^q). \tag{4}$$

$\mathcal{L}_c(\theta)$ is composed of a few terms to ensure the learned system behaves well. The ***gluing loss*** is the core of learning the hybrifold that glued the guard surface in $Z$. Though in each domain the pre- and post-reset state $x^+$ and $x^-$ can be far away from each other, they are encouraged to be close to each other in $Z$. As we do not assume any label other than the time series data, we label the state with large finite time differences $\|\frac{x_{k+1}-x_k}{\Delta t_k}\|_2^2$ as the transitions and denote the indicator set as $\mathcal{C}$. The ***dynamics loss*** is designed to learn the flow in the latent space by matching the prediction of flow with the encoded trajectories on $Z$.

---

**Algorithm 1** CHyLL

**Require:** Trajectory dataset $\mathcal{X}$; curriculum $\{T_1 < \cdots < T_L\}$; steps per length $S$; batch size $B$; decoder steps $K$
1: **for** $\ell = 1, \ldots, L$ **do** // Phase I: Train Encoder & Flow
2:   **for** step $= 1, \ldots, S$ **do**
3:     $\{x_{0:T_\ell}^q\}_{q=1}^B \sim \mathcal{X}$        $\triangleright$ Sample minibatch
4:     $z_{0:T_\ell}^q \leftarrow \Phi_\theta(t, E_\theta(x_0^q))$ $\triangleright$ Rollout in latent space
5:     $\mathcal{L}_d(\theta) \leftarrow$ Equation (3)
6:     $\theta \leftarrow \theta - \eta \nabla_\theta \mathcal{L}_d(\theta)$
7:   **end for**
8: **end for**
9: Flatten $X \leftarrow \mathcal{X}$; // Phase II: Reconstruction
10: **for** $i = 1, \ldots, K$ **do**
11:   $x_{\text{batch}} \sim X$, $x_{\text{noisy}} \leftarrow x_{\text{batch}} + \epsilon$, $\epsilon \sim \mathcal{N}(0, \sigma^2 I)$
12:   $\mathcal{L}_d(\xi) \leftarrow \text{MSE}\big(D_\xi(E_\theta(x_{\text{noisy}})), x_{\text{noisy}}\big)$
13:   $\xi \leftarrow \xi - \eta \nabla_\xi \mathcal{L}_d(\xi)$
14: **end for**

---

As the gluing loss encourages the points at the guard surface that are far away to move close, it is possible that the $\theta$ converges to a trivial solution, where the latent collapses to a single point and the flow becomes static. To mitigate this issue, we have the ***latent collapse loss*** which enforces a minimal covariance for each dimension of the latent features to avoid the trivial solution. Finally, we note that though the Whitney Embedding Theorem suggests the latent is singularity-free, the manifold can still be highly distorted, which makes the decoding and dynamics learning complicated. To avoid this issue, we consider the induced Riemannian metric of $E_\theta(\cdot)$ and enforce the ***conformal loss*** to only scale while preserving the angle for the samples: $(\frac{\partial E}{\partial x})^\top \frac{\partial E}{\partial x}|_{x=x_i} \to \theta_c I_n, \forall x_i$. After optimizing the loss $\mathcal{L}_c(\theta)$, we can train the decoder by simply minimizing the reconstruction loss:

$$\mathcal{L}_d(\xi) = \text{MSE}(D_\xi(E_\theta(x_k^q)), x_k^q). \tag{5}$$

Given that the encoder $E(\cdot)$ is almost injective (except the guard surface), the whole pipeline can be decoder-free by solving the nonlinear projection by, e.g., Levenberg-Marquardt(LM) algorithm, given a sufficiently good initialization. Thus we have the refinement steps for theoretical analysis:

$$\min_x \|E_\theta(x) - z\|_2^2, z \in Z. \tag{6}$$

Now we summarize the training method in Algorithm 1. As predicting the state of a dynamical system over a long duration is difficult, we design a curriculum to learn the flow with increasing horizon length to make the training more stable.

## 6 NUMERICAL EXPERIMENTS

We apply the adjoint method for training using the Neural ODE Chen et al. (2018) in the latent space to rollout the states. To identify the point to glue, we compute the empirical Lipchitz constant for all data point and glue the pairs outside of the two-sigma boundary. The details of the experiment setup are presented in Appendix B. The key questions we want to answer are: **Q1)** Can we learn discontinuous flow of $\mathcal{H}$ without discrete components? **Q2)** Can we identify the topological structure of $\mathcal{H}$ as predicted by the *hybrifold* structure? **Q3)** How does the learned hybrifold structure support the downstream tasks?

### 6.1 LEARNING THE FLOWS

We showcase the proposed algorithm on a few classical examples, including the torus, Klein bottle, and the bouncing ball. Consider a hybrid system with one domain $D_1 = [0, 1]^2$ associated with the

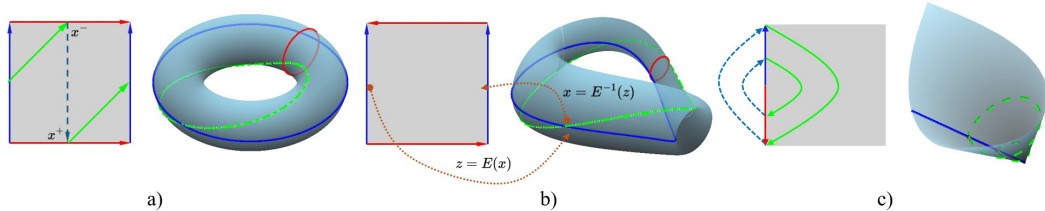

a)                          b)                          c)

Figure 3: Boundaries in the same color are glued following the directions of the arrows. a) Torus and the flow. b) Immersion of the Klein bottle. Two points far away can be close after gluing. c) Inelastic bouncing ball. We see that after gluing, all the hybrid flows become continuous.

constant vector field $\dot{x} = c, \forall x \in [0,1]^2$. The system has one edge $e$ with start and end both on $D$. The guard surface of the system contains two neighboring edges of $D$:

$$G(e) = G_1 \cup G_2, \text{with}, G_1 = \{x|x_2 \in [0,1], x_1 = 1\}, G_2 = \{x|x_1 \in [0,1], x_2 = 1\}. \quad (7)$$

The reset map of **Torus**, i.e., $R_T$ glues the opposite edges without twist, while the reset $R_K$ for the **Klein Bottle** twists one pair of the edge:

$$R_T(x) \sim \begin{cases} (0, x_2) & x \in G_1 \\ (x_1, 0) & x \in G_2 \end{cases}, \quad R_K(x) \sim \begin{cases} (0, x_2) & x \in G_1 \\ (1 - x_1, 0) & x \in G_2 \end{cases}. \quad (8)$$

The **Bouncing Ball** system satisfy the linear dynamics with gravity $g$: $\dot{x}_1 = x_2, \dot{x}_2 = -g$. The guard surface $G_B(e) = \{(x_1, x_2)|x_1 = 0, x_2 \le 0\}$ indicates the ground where elastic collisions happen and is represented by the reset map $R_B$: $R_B(x) \sim (x_1, -\alpha x_2), (x_1, x_2) \in G_B$.

The systems are illustrated in Figure 3 .To learn the dynamics, we compare the proposed method with four method. 1) Neural ODE (Chen et al., 2018), 2) Deep Koopman Operator (Lusch et al., 2018) based on spectral theory, 3) Event Neural ODE (Chen et al., 2020) that contains each component of $\mathcal{H}$, and 4) Latent Neural ODE (Chen et al., 2020) that also learns the latent dynamics but without topological loss. For the latent ODE with autoencoder, the structure is identical as the proposed method but with the decoder trained with the continuous part with only reconstruction loss.

We use an event-based simulation to obtain the dataset $\mathcal{X}$ with 1000 trajectories. We use 800 trajectories for training and 200 for testing. The MSE loss of the testing dataset are listed in Table 1. We can see that the proposed method consistently outperforms the baselines in MSE. The predicted trajectories for the bouncing ball, torus, and the Klein bottles are illustrated in Figure 4, 5, and 6, respectively. With the refinement of the decoder by the Levenberg–Marquardt algorithm, the performance of the proposed method further improves in many cases. In the bouncing ball case, though the Neural ODE can consistently provide good solutions, we observe a large ground penetration, which is not seen in the proposed method. In the torus and Klein bottle example, we see that all the baseline fails to capture the discontinuity in the long term. We note that as the MLP-based event function can hardly satisfy the regularization conditions, they can easily fall in ill-conditions, which is discussed in Appendix A. Finally we note that the latent ODE with autoencoder that learns the continuous and discontinuous parts concurrently also fails to generalize beyond the training horizon.

We further consider a **Three Link Walker** in Figure 20 in the Appendix with nonlinear dynamics and multiple modes, i.e., left stance, right stance, two legs in the air, and slip. We have the full state represented by the joint angle of stance foot, swing foot and torso, i.e., $q := [q_{st}, q_{sw}, q_{to}] \in \mathbb{R}^3$ and its time derivative $\dot{q} \in \mathbb{R}^3$ when the robot does not have slip. The continuous dynamics is the Lagnragian rigid body dynamics $M(q)\ddot{q} + C(q, \dot{q})\dot{q} + G(q) = \pi(q, \dot{q})$ with mass $M(q)$, Coriolis

Table 1: MSE of long-horizon predictions on **four** benchmark systems. The proposed methods achieve the lowest error and remain stable beyond the training horizon, while all other baselines, including Event ODE, either exhibit large errors or fail to converge (*Ill-conditioned*). The method without failure is tested on five independent tests with the mean (std. dev) of MSE. We only show the MSE of one trial for the cases that exhibit an obvious failed pattern.

| | Proposed - Decoder | Proposed - LM | Neural ODE | Latent ODE (RNN) | Koopman | Latent ODE (Auto) | Event ODE |
|---|---|---|---|---|---|---|---|
| Bouncing Ball | 0.271 (0.0720) | 0.237 (0.0629) | 0.332 | 0.710 | 4488 | 1441 | Ill-conditioned |
| Torus | 0.0165 (0.00927) | 0.0164 (0.00989) | 0.0830 | 0.0817 | 7.73 | 48.8 | Ill-conditioned |
| Klein Bottle | 0.0237 (0.00567) | 0.0220 (0.00584) | 0.0627 | 0.0678 | 651 | 38.8 | Ill-conditioned |
| **Three Link Walker** | 0.234 (0.0140) | 0.243 (0.0137) | 0.275 (0.00610) | 0.253 (0.0770) | 151.95 | 0.567 | Ill-conditioned |

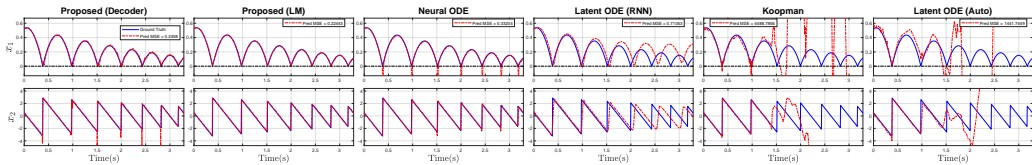

Figure 4: Bouncing Ball. The proposed method does not have penetrations and precisely preserves the change at impact. After refinement by Equation (6), the inaccurate velocity estimation at the impact is eliminated.

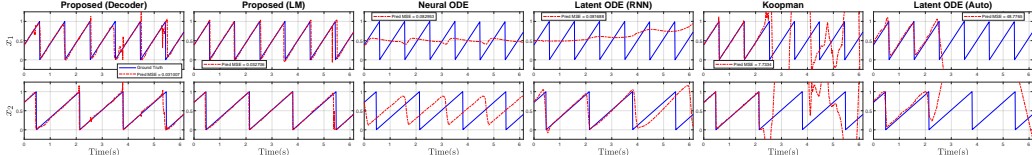

Figure 5: Torus. All the baseline fails to predict the reset beyond the training horizon (2 secs).

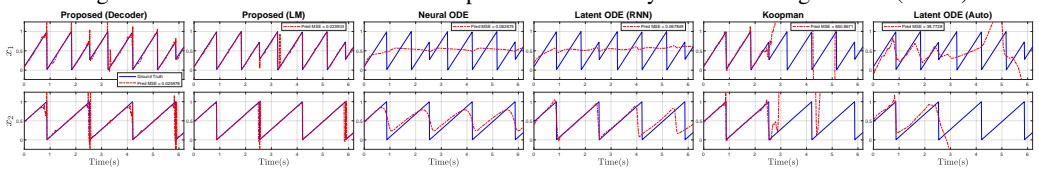

Figure 6: Klein bottle. All the baseline fails to predict the reset beyond the training horizon (2 secs).

matrix $C(q, \dot{q})$, gravity $G(q)$, and a state feedback controller $\pi(\cdot)$ for stable walking. We consider the reset map as the ground impact and relabeling of the stance and swing foot Westervelt et al. (2018):

$$q_{\text{st}}^+ = q_{\text{sw}}^-, q_{\text{sw}}^+ = q_{\text{st}}^-, \dot{q}^+ = \text{Impact}(q^-, \dot{q}^-). \tag{9}$$

We note that the ground impact requires solving a maximal dissipation-based nonlinear programming Halm & Posa (2024). The statistics are presented **in the new line** of Table 1, and we presented the phase portrait in Figure 7 and leg trajectories in Figure 19. We note that in this case CHyLL provides baselines in terms of MSE of prediction, while the baselines either fail to capture the pattern, diverge, or have lower accuracy. This case is trained with 150 step horizon with 500 step prediction.

## 6.2 Topology of the latent and Ablation Study

To verify the structure of the learned latents from the last section, we further apply the persistent homology tool (Tralie et al., 2018) to conduct Topology Data Analysis (TDA). We consider the case with known topology, i.e., bouncing ball, Torus, and Klein bottle, for TDA. Given a point cloud sampled from the latent space, the persistent homology gradually increases the radius of each data point to form simplicial complexes, which generate chains of different dimensions to identify the $k-$dimensional homology group $H_k$. For the chain with different dimensions, i.e., $H_0, H_1, \cdots$, we compute the homology groups, which characterize connected components ($H_0$), loops ($H_1$), voids ($H_2$), and higher-order cavities in the data. As the radius increases, these topological features appear (*birth*) and eventually disappear (*death*). By recording their birth and death scales across dimensions, we obtain persistence diagrams, which summarize the multi-scale invariant topological structure of the data and highlight which features are robust (long persistence) versus noise (short persistence). We then have the Betti number as $\beta = [\beta_0, \beta_1, \cdots]$ that categorizes the data manifold.

We sample a mesh in $x \in \mathbb{R}^2$ and obtain the latent point cloud to compute the Betti number. The persistent diagram of all three cases is illustrated in Figure 8a. For bouncing ball, the $z$ point cloud

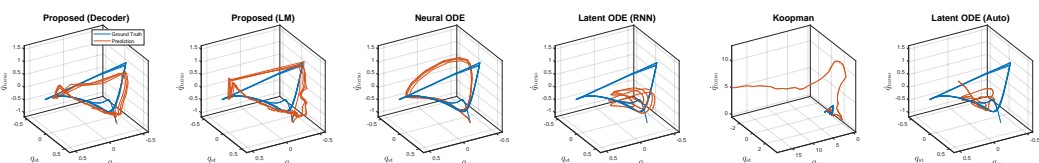

Figure 7: The phase portrait of the trajectories of three three-link walker. We note that the baseline fails to capture the pattern or has a large prediction error. The joint trajectory v.s. time is hown in Figure 19.

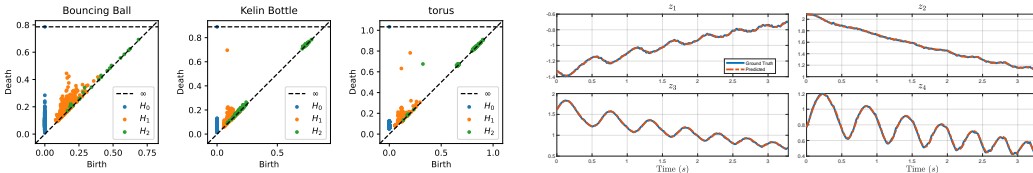

(a) Persistent diagrams of latent point clouds.          (b) Latent trajectories of the bouncing ball.

Figure 8: Persistent homology and latent trajectory analysis. (a) Points far from the "birth=death" line correspond to long-lived holes, which are more likely to be scale-invariant. (b) Latent trajectories reveal smooth dynamics while the $x$-space trajectories have discontinuity. Both (a) and (b) suggest that the latent space is continuous across different scales of data associations.

Table 2: Ablation study of each term in Equation (3). We tested the effect of terms in Equation (3) by disabling one at a time. We show the mean MSE across five trials. The LM refined solution is illustrated in the bracket.

|  | Gluing Off | Conformal Off | Collapse Off | $z \in \mathbb{R}^3$ | $z \in \mathbb{R}^8$ | Proposed |
|---|---|---|---|---|---|---|
| Bouncing Ball | 0.371 (0.336) | 1.99 (2.07) | 0.133 (0.145) | 0.845 (0.858) | 0.195 (0.145) | 0.271 (0.237) |
| Torus | 0.0461 (0.038) | 0.0911 (0.0891) | 0.0134 (0.0127) | 0.0436 (0.0389) | 0.00672 (0.00584) | 0.0165 (0.0164) |
| Klein Bottle | 0.0402 (0.0366) | 0.0951 (1.022) | 0.0373 (0.0339) | 0.102 (0.125) | 0.0171 (0.0154) | 0.0237 (0.0220) |

has a $H_0$ point that has infinite longevity, while $H_1$ and $H_2$ holes vanish quickly after emergence. Thus, we can see that the Betti number is likely to be $\beta_B = [1, 0, 0]$, which corresponds to the projective plane that is consistent with the topological structure of the hybrifold structure as discussed in Simic et al. (2005). Similarly, we can see that the $z$ point cloud of the torus has a single $H_0$ point that lasts forever, two $H_1$ points that live long, and one $H_2$ hole. Thus, the Betti number of this point cloud is $\beta_T = [1, 2, 1]$, also consistent with the Betti number of the Torus. In the Klein bottle, we see that the Betti number of the $z$ point cloud is $\beta_K = [1, 1, 0]$, consistent with the topology of the underlying hybrifold. Finally, we show the latent trajectory of the bouncing ball in Figure 8b. Compared with Figure 4, the latent trajectory is smooth. For more visualization, see Appendix C.

The ablation study about each term of Equation (3) is also reported in Table 2. We consider disabling one term in Equation (3) at a time. We note that the *conformal loss* is the key to improving the performance by reducing the latent distortion. Increasing the latent dimension from 4 to 8 significantly improves the results. When decreasing the latent dimension from 4 to 3, the result also degenerates, which can be explained by the bound indicated in Theorem 2, especially for the Klein bottle that does not admit a singularity-free representation in 3-dimensional space. The gluing loss also improves the performance significantly. We note that the latent collapse loss only improves the performance in the Klein bottle, the topology of which is more complicated due to the gluing method. Future work will take into account this effect to design the threshold of the latent covariance.

### 6.3 HYBRID STOCHASTIC OPTIMAL CONTROL

Finally, we present the algorithm for the stochastic optimal control of hybrid systems. We consider the ball juggling problem, considering the vertical motions of a ball and a force controller paddle. We use MuJoCo to simulate the systems with elastic collisions. The continuous dynamics of the system, with ball position and velocity $(x_b, v_b)$, paddle state $(x_p, v_p)$, controlled by force $f$ are $\dot{x}_b = v_b, \dot{v}_b = -g, \dot{x}_p = v_p, \dot{v}_p = f/m - g$. We consider a low-level PD controller to generate the force $f = K(v_p - a_k)$ with $a_k$ the desired paddle velocity as the action. We collect 1024 trajectories generated by random actions $a_k$ and apply CHyLL to learn the underlying dynamics for control.

As the ball has no energy loss in the flight phase, we consider an energy-based tracking strategy. To avoid the paddle from directly lifting the ball at the desired position, we add a penalty $\rho$ to avoid the paddle from going too high or too low. Thus the optimal control problem can be derived as:

$$\min_{\{a_k\}_{k=0}^N} \quad (0.5 v_{b,k}^2 + g x_{b,k} - E_{des})^2 + \rho(\text{ReLu}(x_{p,k} - x_{p,\max}) + \text{ReLu}(x_{p,\min} - x_{p,k})) \quad (10)$$

We apply the Model Predictive Path Integral (MPPI) (Williams et al., 2017) control that rolls out the neural dynamics. We consider the same control objective Equation (10) for our method, a Neural ODE-based model, and a Deep Koopman Operator-based model. As all the methods are trained in continuous time, we roll out the dynamics using the RK4 integration scheme. We consider stabilizing the ball at the height of $1.2m$ and $x_{p,\max} = 0.8$. We consider the same planning horizon and control rate for all the methods and repeat the control for 10 trials and present the result in Table 3. We find that Koopman operator outperforms the proposed method in terms of the mean tracking

cost and has compatible standard deviations. The Neural ODE fails in this task, possibly due to the ground penetrations that result in inaccurate collision point predictions.

Table 3: Tracking cost statistics of MPPI for ball juggling with planning horizon $0.8s$ with 40 HZ control rate.

|  | 1 | 2 | 3 | 4 | 5 | 6 | 7 | 8 | 9 | 10 | Mean | Std. Dev |
|---|---|---|---|---|---|---|---|---|---|---|---|---|
| Proposed | 10.49 | 4.13 | 5.67 | 3.83 | 3.94 | 5.21 | 4.08 | 3.64 | 7.41 | 3.55 | 5.195 | **2.10** |
| Koopman | 8.48 | 7.79 | 1.44 | 3 | 5.56 | 2.88 | 2.15 | 2.71 | 3.72 | 6.19 | **4.392** | 2.33 |
| Neural ODE | 291.77 | 300.05 | 280 | 286.85 | 203.16 | 249.62 | 223.22 | 225.66 | 254.77 | 274.33 | 258.943 | 31.35 |

## 7  DISCUSSIONS

**Role of loss terms**: We find that the conformal loss plays an important role in preventing distortions and ensuring the decoding quality. As discussed in Cohn et al. (2022), a single MLP-based decoder may have singularities and does not preserve the topological structures. After adding the conformal loss, the decoding quality greatly improves even with a single MLP as the decoder. Without this loss, atlas learning with multiple networks in different charts has poor performance. The covariance threshold for the anti-collapse loss needs further tuning in the case with a simpler gluing pattern, as we observe that it does not always improve the performance.

**Integration in the latent space**: We roll out the trajectories in the latent space via the Neural ODE. As the latent space $\mathbb{R}^m$ is the ambient space, the current integration scheme is not intrinsic on the manifold. We note that this effect contributes to the degraded LM performance if the latent error is large, which means the rollout largely deviates from the manifold. To mitigate this issue, one can consider i) add additional constraints or ii) conduct the intrinsic integrations scheme in a moving charts Lou et al. (2020). We note that i) may require solving differential algebraic equations (Koch et al., 2024) or variational integrations Saemundsson et al. (2020), which is computationally expensive, and ii) requires the Riemannian exponential or logarithmic of the induced manifold of encoder $E_\theta(\cdot)$. These implementations can be future work to address this potential failure mode.

**Comparison with Koopman operator**: We find that though the long-term performance of the deep Koopman operator is worse than the proposed method, the short-term performance is compatible. In the controlled bouncing ball experiment, we note that the continuity of the flow also depends on the external input. In this case, the gluing detection needs to distinguish the discontinuity introduced by the guard/reset or control input. Future work will need to carefully distinguish different sources of discontinuity to further improve the performance of the propose method in the controlled systems. Exploring the connection between the proposed embedding theorem-based method and the Koopman operator based on the spectral theorem is an interesting future work. One possible solution is to enforce the gluing loss to have a continuous latent in the Koopman feature space, while the alternative is to increase the latent dimension in CHyLL to decrease the nonlinearity, which can be inferred by the bound for the isometry embedding as shown in the Nash Embedding Theorem. The latter one is already shown in the ablation study with an improvement in the performance. Future work will also consider the existing literature on deep Koopman operator, such as Han et al. (2020); Jeong et al. (2025); Kostic et al. (2024).

**Stochasticity and partial observability**: In real-world scenarios, we note that the time series data is generally generated by stochastic hybrid systems (sensor noise, stochastic reset/guard), and sometimes only partial observation of the system is accessible. In the stochastic setting, we need to reformulate the problem for this stochastic hybrid system setting Tejaswi et al. (2025). For a partially observable system, the internal dynamics of the unobserved subspace need to be further taken into account Westervelt et al. (2003). Though these settings are beyond the scope of this work, they are interesting future work that is worth exploring for real-world applications.

## 8  CONCLUSIONS

In this work, we presented the CHyLL (Continuous Hybrid System Learning in Latent Space) to learn the dynamics of hybrid systems from only time-series data without trajectory segmentation, event functions, or mode switching. The insight of CHyLL is to reformulate the state space of the hybrid system as a continuous quotient manifold glued by the reset map, and then learn the flow on it. CHyLL learns the glued manifold and the latent vector field concurrently via a topology-inspired loss function. We show that CHyLL can accurately recover the flow of hybrid systems, where the other method fails. By the topological data analysis, we further showcase that CHyLL is capable of identifying the topological structure of the learned quotient manifold. Finally, we applied the proposed method in downstream task like stochastic optimal control.

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

APPENDIX

## A    REGULARIZATION CONDITION OF $\mathcal{H}$

We introduce several regularization conditions of $\mathcal{H}$ that define the well-behaved hybrid system and account for some difficulties in the Event Neural ODE. This part directly adapts from the assumptions shown in (Simic et al., 2005).

A1  $\mathcal{H}$ is deterministic and non-blocking.

A2  There exists a $d$ such that each domain $D_i$ is a connected $n$-dimensional smooth submanifold of $\mathbb{R}^d$, with piecewise smooth boundary. The angle between any two intersecting smooth components of the boundary is nonzero.

A3  Each guard is a smooth $(n-1)$-dimensional submanifold of the boundary of the corresponding domain. The boundary of each guard is piecewise smooth (or possibly empty).

A4  Each reset is a diffeomorphism from its domain $G(e)$ onto its image. The image of every reset lies on the boundary of the corresponding domain. Moreover, if $e = (i, j) \in E$ and $R_e(p) = q$, then $V_i(p) = 0$ if and only if $V_j(q) = 0$.

A5  Elements of $\overline{\mathcal{G}} \cup \overline{\mathcal{R}}$ (i.e., sets which are closures of guards and images of resets) can intersect only along their boundaries. Furthermore, if $p \in \bar{G} \cup \bar{R}$, then $p$ can be of only one of the following four in (Simic et al., 2005).

A6  For all $e = (i, j) \in E$, the following holds: on $\operatorname{int} G(e)$, $V_i$ points outside $\operatorname{int} D_i$; on $\operatorname{int}(\operatorname{im} R_e)$, $V_j$ points inside $\bar{D}_j$.

A7  Each vector field $V_i$ is the restriction to $D_i$ of some smooth vector field, which we also denote by $V_i$, defined on a neighborhood of $D_i$ in $\{i\} \times \mathbb{R}^n$. Each reset map $R_e$ extends to a map $\tilde{R}_e$ defined on a neighborhood of $\overline{G(e)}$ in $D_i$ such that $R_e$ is a diffeomorphism onto its image, which is a neighborhood of $\operatorname{im} R_e$ in $D_j$.

A8  If $p \in D_i$ is on the boundary of $D_i$ and $V_i(p)$ points inside $D_i$ then $p$ is in the image of some reset.

These regularization conditions avoid pathological behaviors in learning, such as: A1) ensures $\mathcal{H}$ has unique solutions; A3) ensures $\mathcal{H}$ does not have infinitely many state changes at a single time; A6) ensures the trajectories will not go back and forth around the guard surface.

We note that learning each component of $\mathcal{H}$ explicitly using Event Neural ODE or its discrete variants generally does not have guarantees on the satisfaction of these conditions. According to our experience, it is easy to have infinitely many state changes that stall the training process.

## B    EXPERIMENTAL SETUP

We emphasize reproducibility and transparency. All code, configuration files, and preprocessed data used in these experiments will be released under an open-source license upon publication.[4] The following paragraphs describe the model architecture, hyper-parameters, and training schedule in sufficient detail to enable exact replication.

### B.1    MODEL ARCHITECTURE AND PARAMETERS

All neural components are implemented as multilayer perceptrons (MLPs) with ReLU activations. Table 4 summarizes the hidden-layer configurations for the event-driven Neural ODE (vector field, guard, and reset functions). Table 5 summarizes the parameters for latent Neural ODE with RNN encoders. Table 6 details the full parameterization for every baseline and ablation, including our proposed method, continuous ODE variants, and Koopman baselines.

---

[4]The repository URL will be provided in the camera-ready version.

Table 4: Hidden layers for each component of the event-driven Neural ODE. Brackets indicate layer width; "$\times k$" denotes $k$ identical layers.

|  | Vector Field | Guard | Reset Function | Vector Field Dim. |
|---|---|---|---|---|
| Event ODE | [64]$\times$2 | [64]$\times$3 | [64]$\times$3 | 2 |

Table 5: Hidden layers for each component of the ODE (RNN).

|  | Vector Field | Encoder | Decoder | Vector Field Dim. |
|---|---|---|---|---|
| ODE (RNN) | [100]$\times$2 | GRU(100) | Linear$\times$3 | 20 |

## B.2 TRAINING SCHEDULE

We adopt a curriculum strategy in which the rollout horizon grows from 10 to 200 steps according to the sequence $\{10, 20, 40, 80, 150, 200\}$. Each horizon is trained for 2 000 gradient-descent updates, except the final 200-step stage, which is trained for 4 000 updates to ensure stability at long time horizons. The trajectory rollout batch size is fixed at 4 096 throughout all phases. Unless otherwise stated, all models are optimized with the same learning-rate schedule and weight initialization to ensure fair comparison.

This level of specification, combined with our forthcoming open-source release, is intended to make the reported results fully reproducible and to facilitate direct benchmarking by the research community.

## C MORE VISUALIZATIONS

### C.1 LATENT TRAJECTORIES

The latent trajectories for the Klein bottle and the torus are presented in Figure 9 and 10. We see that both trajectories are continuous even the original flows are highly discontinuous.

More trajectories for the Torus, Klein bottle, and bouncing balls are shown as follows:

## D ILLUSTRATION OF THE THREE-LINK WALKER

Table 6: Model parameters for all other methods. "N/A" indicates the component is not used.

|  | Proposed | ODE | Koopman | ODE (Auto) |
|---|---|---|---|---|
| Vector Field | [64]$\times$2 | [128]$\times$2 | Linear | [64]$\times$2 |
| Encoder | [64]$\times$3 | N/A | [64,128,256] | [64]$\times$3 |
| Decoder | [128]$\times$8 | N/A | [256,128,64] | [64]$\times$3 |
| Vector Field Dim. | 4 | 2 | 256 | 4 |

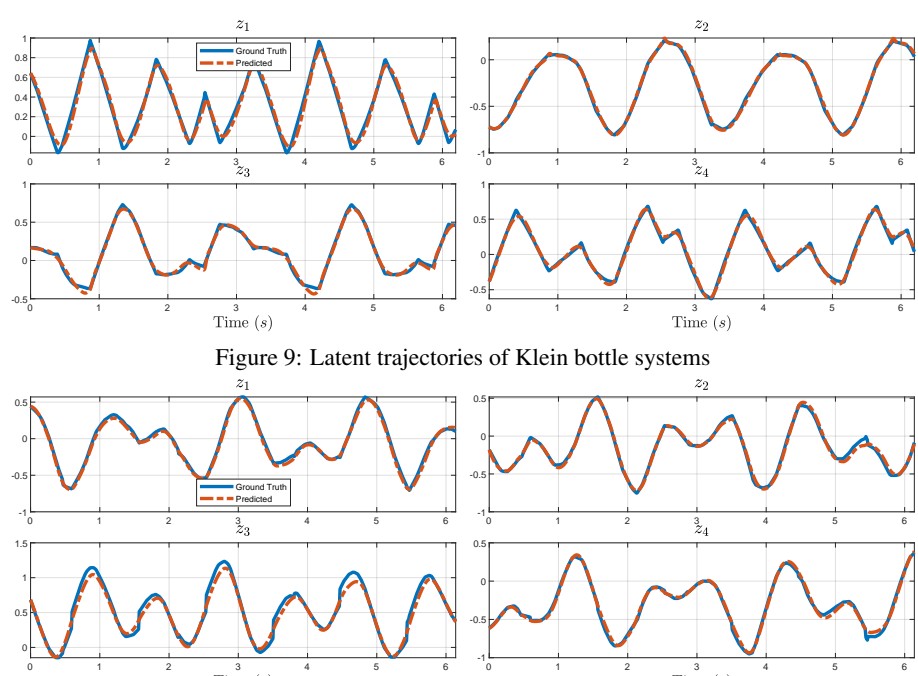

Figure 9: Latent trajectories of Klein bottle systems

Figure 10: Latent trajectories of torus systems

Figure 11: Bouncing ball trajectories

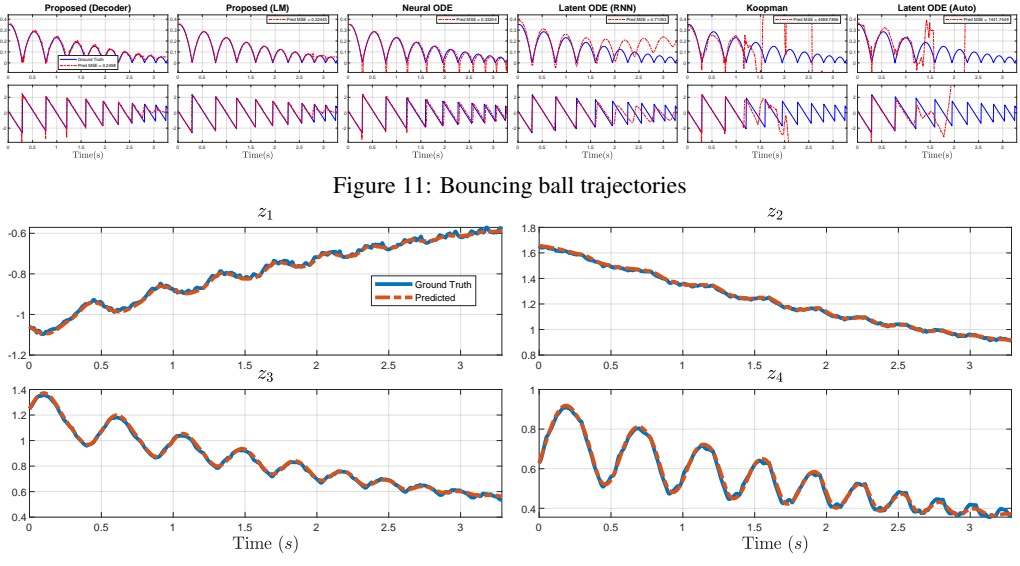

Figure 12: Latent of bouncing ball by CHyLL

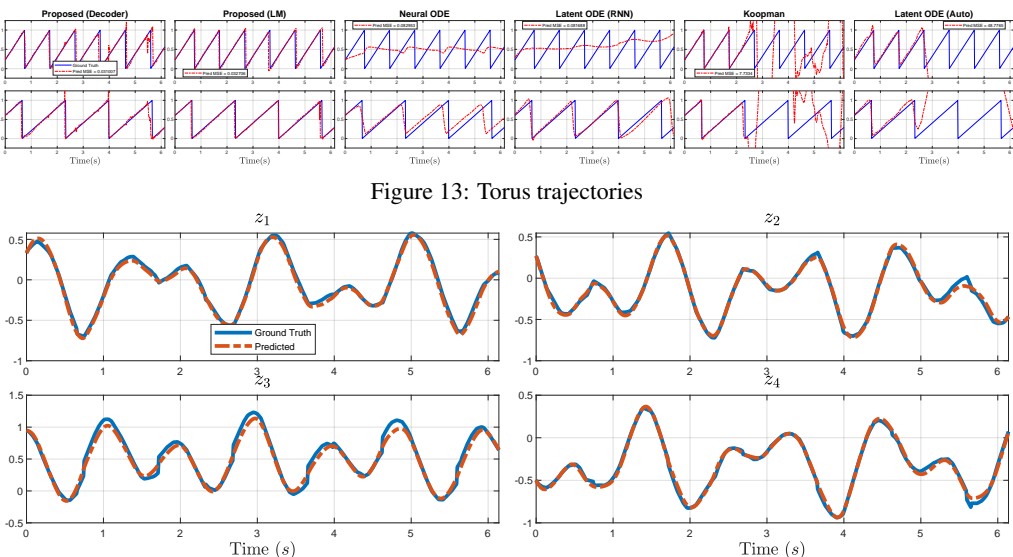

Figure 13: Torus trajectories

Figure 14: Latent of torus by CHyLL

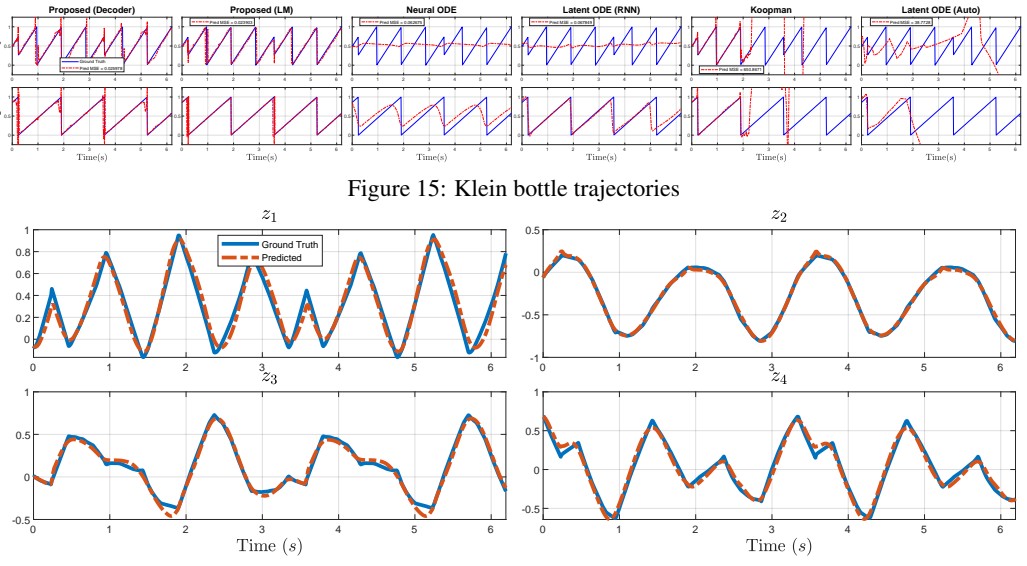

Figure 15: Klein bottle trajectories

Figure 16: Latent of Klein bottle by CHyLL

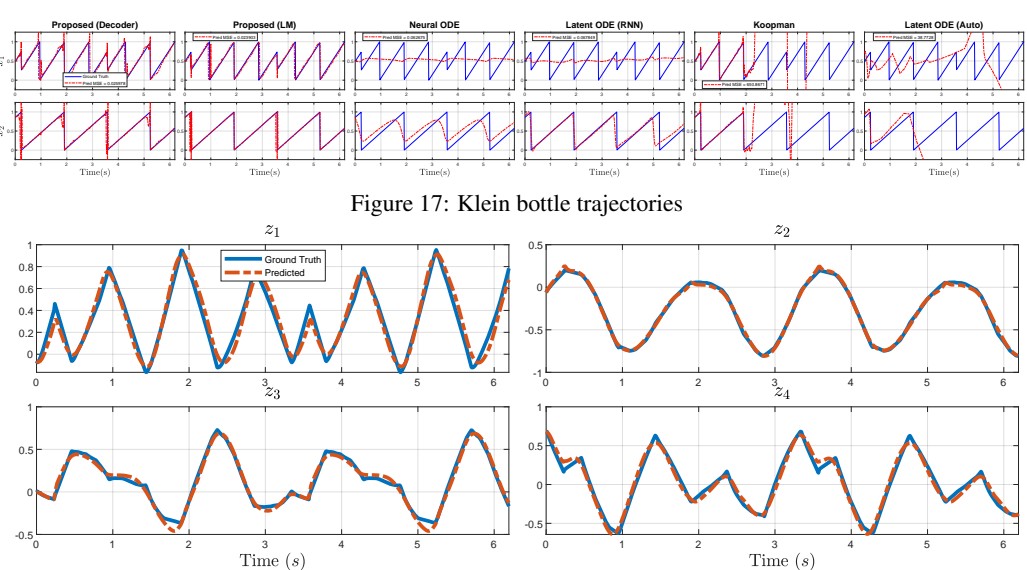

Figure 17: Klein bottle trajectories

Figure 18: Latent of Klein bottle by CHyLL

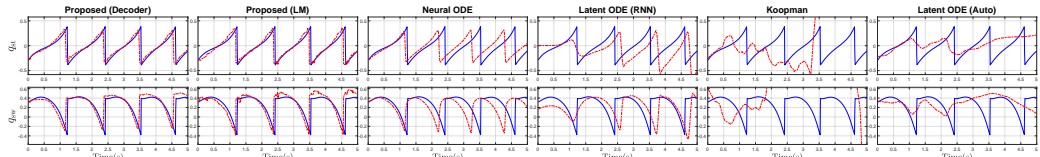

Figure 19: Comparison of the prediction of stance and swing foot joint angle.

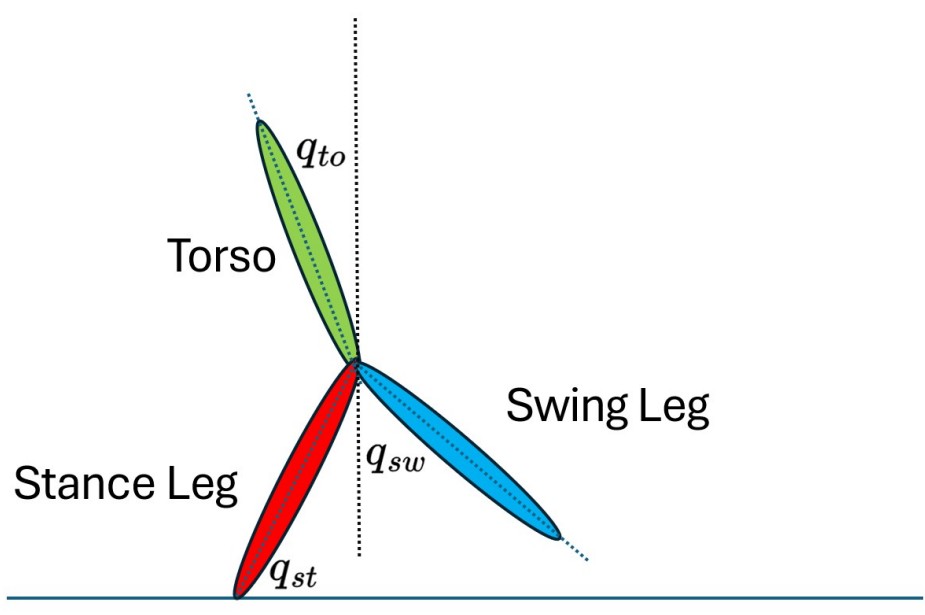

Figure 20: The model of the three-link walker. The control inputs are mapped to the joints via rigid body dynamics in generalized coordinates.

