# OpenReview forum: "CHyLL: Learning Continuous Neural Representations of Hybrid Systems"
_ICLR.cc/2026/Conference — ICLR 2026 Conference Withdrawn Submission_

### Official Review · Reviewer_wC3u · 2025-10-24

**Soundness:** 3
**Presentation:** 2
**Contribution:** 1
**Rating:** 2
**Confidence:** 2

**Summary:**

The authors propose chyLL, a method to learn continuous representations of hybrid systems in latent space. The method ensures the representations become continuous in a higher-dimensional latent space, then fit a latent ode there and decode back, avoiding mode labels, event functions, or segmentation. The contribution is the gluing/conformal/anticollapse losses to learn this manifold from time-series alone, with demos (bouncing ball, torus/klein bottle topology checks, and a control task).

**Strengths:**

The paper is clear and well-motivated, and the pipeline and objectives are easy to follow.
The route explored is worthwhile as a method to modeling hybrid dynamics that generalizes across systems.
The proposal of learning a glued quotient manifold for hybrids via gluing + conformal losses is nice, and the training setup seems sensible (e.g. curriculum, anti-collapse, LM projection).

**Weaknesses:**

1. Although latent encoding for ODEs is not a new avenue, the quotient/gluing idea is a nice addition, but as the authors hint at in the paper, learning the correct 'glued' space might be hard in principle, and the lack of guarantees can be concerning.

2. The experimental scope is a little narrow, with toy problems/examples, and only a few comparative methods. The results for the ball juggling with MPPI experiment also show a deep Koopman baseline achieving a lower mean tracking cost, without much of a detailed explanation.

3. It's hard to comment on the scalability and robustness of the approach since there are no results under sensor noise/partial observability, many-guard/mode systems, or higher-dimensional robotics experiments.

**Questions:**

2. The authors should expand further on the limitation of the proposed method, and its failure mode, with respect to existing literature.

1. Can you provide any diagnostic or bound (e.g., jump norms across detected guards, encoder/decoder Jacobian conditioning near resets) that indicate the learned latent is truly continuous, or conditions under which it fails?

2. How sensitive is performance to the gluing/conformal/anti-collapse weights and latent dimension? It would be valuable to include a sweep or at least failure modes when turning each off.

3. Can you expand further on why deep Koopman wins on mean cost, and whether CHyLL improves with different horizon/MPPI settings or controller?

4. How do you think the method would behave under more realistic sensor noise or partial observability?

---

> ### Author Response · Authors · 2025-11-24
> **Responses (part 1)**
>
> We thank the reviewer for the constructive feedback and great suggestions! We are sorry that this reviewer gave a score of 2 in the first round, but we wish our revision could address all the issues.
>
> We here include:
> 1. Clarification of the novelty.
> 2. New experiment on a robotic system in the Table. 1 and Fig. 7.
> 3. More discussions on stochasticity and partial observability.
> 5. Ablation studies in Sec. 6.2 and Table. 2.
> 6. Responses to each question
>
> Weakness:
>
> **1. Math rigor and guarantees:**
> The existing method for learning hybrid systems needs to assign each segment of this trajectory to a different domain, which is exponentially hard. While our method fundamentally solved this problem by "gluing". This transformation is not possible without the topology/geometry-inspired loss design.
>
> We mentioned the **numerical** difficulty of learning the correct "glued" space. However, our paper **successfully** mitigate this issue by integrating the **Embedding theorem** to numerically represent the manifold indiced in (Simic 2025) with only proof of existence.
>
> As shown in Theorem 1, the continuous structure for hybrid system exist; and by Theorem 2 and Remark 2, the continuous structure can be **parameterized without discontinuity/singularity** by simply increasing the latent dimension. All the loss function design are based on these theory, which has **provable guarantees** up to the assumption that illustrated in Appendix. A.
> To the best of our knowledge, there is not such a prior work that address this problem to actually have this hybrifold geometrically constructed **purely from time series data**.
>
> To the best of our knowledge, this is the first demonstration that the topological hybrifold structure can be **learned** and **realized geometrically** in a latent space, enabling continuous flow learning for hybrid systems. We are grateful that **Reviewer 488A** also highlighted this point.
>
> **2. Experiment Scope:**
> The main goal of this work is to verify the "gluing" idea for learning a hybrid system. Thus, we have to test the method on system with **known** topology.
>
> We also added one more experiment on a high-dimensional robot (6 DOF state space) with multiple modes, as detailed below, to show further possibilities for scaling this method up.
>
> **3. Scalability, robustness, new experiment:**
>
> The questions regarding the partial observation and noise are answered in responses to Q6. These questions are related to the internal dynamics or stochastic hybrid systems that are beyond the scope of this work. They are excellent questions worth discussing in future work.
>
> To better illustrate CHyLL within the appropriate setting, we have added additional deterministic simulations, a three-link walker with 6 DOF state space and multiple modes (left stance, right stance, flying, slip...).
>
> We show that the CHyLL outperforms all Koopman, auto-encoder-based, and RNN-based latent ODEs. CHyLL also consistently outperforms ODE in 5 repeated experiments in terms of the MSE. The new experiments suggested that CHyLL can be extended to more complicated scenarios in the future.
>
> For methods without a failed pattern, we tested them on five different trials and reported the MSE(std.dev of MSE) in Table 1. The revised manuscript:
>
> i. CHyLL:** Decoder: 0.233(0.0134) | LM: 0.243(0.0137)
>
> ii. Neural ODE: 0.275(0.0610)
>
> iii. RNN-ODE: 0.253 (0.0770)
>
> iv Koopman (Diverge): 151.9
>
> v. AutoEncoder (Diverge): 0.567
>
> vi. Event ODE (Ill-conditioned)
>
> This example further showcases that CHyLL can handle high-dimensional cases with complex nonlinear dynamics and impact. For even higher-dimensional legged robots, one future direction is to combine with model-based inductive biases to further reduce the nonlinearities. **The result is presented in new line in Table. 1 and Figure. 7.**

---

> ### Author Response · Authors · 2025-11-24
> **Responses (part 2)**
>
> **Q2. Failure mode:**
> In terms of computation, the main bottleneck of this work is the distortion of the glued space. One intuition is that when gluing the paper to formulate a cylinder, the paper can be folded or stretched in an arbitrary way (topology does not care about the shape). In this case, the encoder will need to map the state to a highly nonlinear latent space that is hard to decode. This is why we introduced the conformal loss. **In the ablation study**, we see that turning off the gluing loss results in lower accuracy.
>
> Another limitation is the integration scheme; we note that the latent ODE is not intrinsic, i.e., $\hat{z}$ can leave the latent manifold, i.e., $\mathrm{Img} E(\cdot)$. When leaving the manifold, the decoder/LM performance will also degrade. An intrinsic integrator or variational integrator is a possible solution to this problem. The solution to this problem is also included in the discussion section.
>
> Compared to the existing literature, the main contribution of this work is to avoid mode selection that is exponentially hard. We only require the time series data and compute the empirical Lipschitz continuity to achieve this goal. To the best of our knowledge, this is the first work that achieves this goal by combining topology and the embedding theorem in this learning framework. The existing literature is learning in the original space that still has the discontinuity.
>
>
> **Q3. Disgonositc or Bounds**
>
> The latent flow is plotted in Fig. 8(b) and the Appendix to compare with the flow in the original state space to show this effect. We find that the latent flow is indeed continuous. The discontinuity of the decoder can be verified by the fact that the latent flow is continuous while the decoded flow is not.
>
> We have computed the **Lipschitz constant/jump norm** of the flow (both in M and Z space) here. For the bouncing ball case, the original mean Lipshitz constant of a trajectory is 18.7857 with a maximal value 592.6. While in the latent space the mean is **1.63** and max value is 7.0. The deviation from mean value changes from 30x to 3x is a sign that the gluing method works.  For Klein bottle, the changes from M to Z space is 3.1(140) to 3.4(6.9). The deviation changes from 45x to 2x. For Torus, the change is 2.64(106) to (3.7)28. The deviation changes from 50x to 9x. All changes suggested that the latent flow indeed becomes continuous after gluing.
>
> We also noted that the **Topological Data Analysis (TDA)** in **Sec. 6.2** is the **best/standard** way to showcase the continuity. Fig. 8(a) (persistent diagram) is the standard tool in TDA to classify the manifold based on how points are connected. It showed the longevity of the topological invariants of the latent point cloud, which suggested that there indeed exist such holes in the latent manifold across different scales of data association.
>
> For a discontinuous latent space, the invariants will not exist due to the large gap around the boundary. For latent dimensions higher than 3, this is the most reliable way to show the continuity without visualization.
>
> One failure case we can mention is that when disabling the gluing loss, the TDA fails to provide the desired Betti number, as the latent point clouds are disconnected.
>
> **Q4. Ablation:**
> We have included a ablation study in **Sec 6.2** and **Table. 2** to show the effect of the different losses and dimension. We find that the dimension, gluing, and conformal loss are the key to having a good prediction result. The latent collapse loss is important when the space is highly distorted.
>
> We find that decreasing the dimension from 4 to 3 makes the prediction of the bouncing ball, torus, and Klein bottle degenerate a lot. As a three-dimensional parameterization can not embed a Klein bottle without a singularity or self-intersection, this is a fundamental limit. When increasing the latent dimension from 4 to 8, we see that the performance improves a lot. The improvement can be explained by the **Nash Embedding Theorem** that higher dimension space ensures a possible **isometric embedding** that has little distortion to allow better performance on encoder/decoder.
>
> Similarly, the conformal loss that avoids the distortion is also the key to better results.
>
> The collapse loss is key to the Klein case, as the gluing is more complicated (twist one pair of opposite edges). This finding is key to helping us tune the parameters for better results.

---

> ### Author Response · Authors · 2025-11-24
> **Responses (part 3)**
>
> **Q5. Deep Koopman:**
>
> We note that in this case, we have a control input that makes the gluing point detection inaccurate. As we are using the empirical Lipschitz constant to compute the gluing point, the discontinuity introduced by the external control will corrupt the detection. While the controlled Koopman operator does not have this issue.
>
> In our experiment, we find that increasing the planning horizon does not help MPPI/Koopman, as imperfect learned dynamics introduce additional regret (difference between the ideally best and actual cost) in Model Predictive Control, which is more obvious with longer time horizon. In the MPPI-based world model, the planning horizon is usually less than 3 steps, with the Q function to avoid this issue, while we use dynamics with more than 20 steps.
>
> We consider detecting the accurate gluing point for **controlled** hybrid systems as a future work and we will discuss this in the revised version.
>
> **Q6. Partial observation and sensor noise:**
> We agree that these scenarios with sensor noise or partial observation are important in real-world applications. However, we noted that these scenariors are fundamentally harder in **hybrid systems** setting, which is beyond the scope this work. We have added these points to the discussions.
>
> **Partial observability**: For a hybrid system with state $x$ but only a subspace/submanifold $y=h(x)$ ($\mathrm{dim} y < \mathrm{dim}x$) can be observed, the internal dynamics (the space outside of the set defined by $y=h(x)$) is fundamentally unknown when only time series date is known. This means that same measurements $y$ can **not** uniquely determine the state $x$. In this case, the gluing with partial information is **insufficient** to determine the guard/reset. The partial observation is widely discussed in nonlinear control, such as the classical paper about this effect:
>
> Eric R Westervelt, Jessy W Grizzle, and Daniel E Koditschek. Hybrid zero dynamics of planar biped walkers. IEEE transactions on automatic control, 48(1):42–56, 2003
>
> Though partial observation is an important effect, we note that correctly formulating this problem and discussing it is beyond the scope of this work. However, we will introduce this point in the discussion, and we believe this suggested area is an important future direction.
>
> **Stochasticity**: Similar to the discussion with review **EUQm**, we note that the hybrid system becomes **Stochastic Hybrid System** (SHS) when introducing the noise. In this case, the reset/guard all becomes stochastic function (e.g., whether driver decides should take over the car can be modeled by Hidden Markov Model). In this sense, the gluing method we propose should be re-formulated in this probabilistic setting. This formulation differs substantially from the deterministic hybrid systems that our method is designed for, and for which the underlying theoretical results (e.g., deterministic guards/resets and the existence of a deterministic hybrifold structure) apply.
>
> Because CHyLL relies on learning a continuous embedding of a deterministic quotient structure, applying it directly to SHS would require generalizing the gluing mechanism to a probabilistic setting, e.g., defining probabilistic guard surfaces or uncertainty-aware latent transitions. This is an interesting research direction, and we have added a brief discussion in the revision, but it is outside the scope of the deterministic formulation studied in this paper.

---

> > ### Comment · Reviewer_wC3u · 2025-11-25
> >
> > The authors have addressed my main concerns, I have raised my score accordingly.

---

### Official Review · Reviewer_EUQm · 2025-10-29

**Soundness:** 2
**Presentation:** 3
**Contribution:** 2
**Rating:** 6
**Confidence:** 3

**Summary:**

CHyLL presents a groundbreaking method for learning hybrid system dynamics directly from time-series data, eliminating the need for mode switching or event detection. Its core innovation lies in reformulating the discontinuous state space into a continuous quotient manifold—the hybrifold—by topologically gluing guard surfaces via reset maps. A dual-phase training strategy separately optimizes the continuous encoder/flow and the discontinuous decoder. Evaluations on systems like the bouncing ball and torus show CHyLL's superior prediction accuracy and its ability to identify correct topological invariants.

**Strengths:**

This article is quite abstract, drawing on profound mathematical theories to guide the methodology design. It addresses a very practical problem and has achieved promising results in the selected experimental examples.

**Weaknesses:**

1. Although this article cites many theorems and employs sophisticated mathematical frameworks, its contributions are primarily concentrated on methodological design, with the theoretical contributions not being sufficiently sound. Perhaps the authors could further incorporate theoretical analysis or proofs regarding their methods (such as the design of the loss Eq. 4).
2. The experiments in the paper all use simulated data from simple examples. There is an absence of benchmarking on publicly available real-world datasets.

**Questions:**

1. Although profound mathematical theorems are cited in the paper, I have concerns regarding their rigor. For instance, Theorem 1 states that the manifold is piecewise smooth, while Theorem 2 requires the manifold to be C^r—what is the relationship between these two conditions? Additionally, I seem to be unable to find the precise definition of 'piecewise smooth'.
2. Could there be performance evaluations on some real-world public datasets?

---

> ### Author Response · Authors · 2025-11-24
> **Responses (part 1)**
>
> We would like to thank the reviewer for the constructive feedback and suggestions. We appreciate the reviewer’s comments and have clarified the points accordingly.
>
> We here include:
> 1. New experiment on a robotic system in the Table. 1 and Fig. 7.
> 2. Ablation studies in Sec. 6.2 and Table 2 for analysis of loss design.
> 4. More footnotes in the manuscripts for clarity and rigor of math.
> 5. Responses to each question
>
>
> **Novelty of the method:**
> We appreciate the reviewer’s comments regarding the theoretical component of the work.
> Our contribution does not lie in providing new topological theorems, but in **operationalizing** existing results—particularly the hybrifold construction of Simic et al. (2005) and the Whitney embedding theorem—into a **numerically learnable representation** suitable for modern learning pipelines.
>
> Simic et al. (2005) establish only the **existence** of a continuous (piecewise-smooth) quotient manifold for deterministic hybrid systems. However, their result is purely topological: it does not provide a **computable** or **geometric** realization of the quotient, nor a way to obtain an embedding suitable for numerical learning. Consequently, existing gluing-based approaches in control (e.g., Burden et al., 2015) still rely on *explicit discrete-mode information*, which preserves the discrete structure rather than eliminating discontinuities.
>
> To our knowledge, this is the first demonstration that the topological hybrifold structure can be **learned** and **realized geometrically** in a latent space, enabling continuous flow learning for hybrid systems. Reviewer 488A also highlighted this point, and we have clarified it further in the revision.
>
>
> **Analysis on the method and new ablation study**
>
> To make the design of the loss function clearer, we added an ablation study to showcase the role of each loss. The design criteria of each loss are also highlighted in Section 5.3 to show how we convert the embedding theorem to a computable loss function.
>
> In brief, the gluing loss is inspired by the **topological** equivalence relation, which means the pre- and post- reset state should be identical, as they are temporally close. On the **geometric** side, the conformal loss and collapose loss is designed to avoid the 1) ill-condition / high curvature ofthe  encoder 2) latent collapse to a trivial solution.
>
> In Table. 2, we show that turning off gluing / conformal loss leads to significant performance degradation. Latent collapse loss improves the performance in the Klein bottle, which has a more complicated topology. Future work takes this effect into account to tune the threshold for the latent covariance.

---

> ### Author Response · Authors · 2025-11-24
> **Responses (part 2)**
>
> **Public dataset:**
> We thank the reviewer for the suggestion of evaluating CHyLL on public benchmarks. Most publicly available datasets involving hybrid behaviors (e.g., driving or biological processes) are modeled as stochastic hybrid systems (SHS), where both guards and resets are **stochastic**. This formulation differs substantially from the deterministic hybrid systems that our method is designed for, and for which the underlying theoretical results (e.g., deterministic guards/resets and the existence of a deterministic hybrifold structure) apply. One example of SHS is the driver decides when to overtake the leading car, which is usually modeled by a probabilistic model.
>
> Because CHyLL relies on learning a continuous embedding of a **deterministic** quotient structure, applying it directly to SHS would require generalizing the gluing mechanism to a probabilistic setting, e.g., defining probabilistic guard surfaces or uncertainty-aware latent transitions. We have added a this in the new discussion subsection, while we also note that it is outside the scope of the deterministic formulation studied in this paper.
>
> The existing benchmark for deterministic hybrid systems is mostly in the formal verification community, which relies on known models in simulations.
>
> **New experiment:**
> To better illustrate CHyLL within the appropriate setting, we have added additional deterministic simulations, a three-link walker with 6 DOF state space and multiple modes (one leg stance, two leg stance, flying, slip...).
>
> We show that the CHyLL outperforms all Koopman, auto-encoder-based, and RNN-based latent ODEs. CHyLL also consistently outperforms ODE in 5 repeated experiments in MSE. The new experiments suggested that CHyLL can be extended to more complicated scenarios in the future.
>
> For methods without a failed pattern, we tested them on five different trials and reported the MSE(std.dev of MSE) in Table 1. The revised manuscript:
>
> i. CHyLL:** Decoder: 0.233(0.0134) | LM: 0.243(0.0137)
>
> ii. Neural ODE: 0.275(0.0610)
>
> iii. RNN-ODE: 0.253 (0.0770)
>
> iv Koopman (Diverge): 151.9
>
> v. AutoEncoder (Diverge): 0.567
>
> vi. Event ODE (Ill-conditioned)
>
> We can see that the proposed CHyLL outperforms the other methods in accuracy and capturing the correct pattern. **The result is presented in new line in Table. 1 and Figure. 7.**
>
> **Rigor about math:**
> We thank the reviewer for raising these points.
>
> **Relationship between Theorem 1 and Theorem 2.**
> Theorem 1 establishes the existence of a continuous (piecewise-smooth) hybrifold representation for deterministic hybrid systems.
> Theorem 2, the $C^{r}$ Whitney Embedding Theorem, provides conditions under which such a manifold—when equipped with a $C^{r}$ structure—admits a smooth embedding into a Euclidean space.
> The $C^{\infty}$ case required in Theorem 1 is naturally included as a **special instance** of Theorem 2.
>
> **Meaning of “piecewise smooth.”**
> By “piecewise smooth,” we refer to manifolds that are smooth except on a measure-zero set (e.g., lower-dimensional edges/corners introduced by gluing operations).
> These non-smooth sets do not affect the global continuity guaranteed by Theorem 1 and elaborated in Remark 2.
>
> We have added **three more footnotes** to make these notion explicit and updated the remark title to emphasize the continuity property.

---

### Official Review · Reviewer_488A · 2025-10-31

**Soundness:** 4
**Presentation:** 3
**Contribution:** 4
**Rating:** 6
**Confidence:** 4

**Summary:**

The paper studies the problem of learning hybrid dynamical systems, that is the systems combining continuous flows and discrete mode switches. Diversly from existing approaches, authors propose method CHyLL (Continuous Hybrid System Learning in Latent Space) that   learns directly from time-series data avoiding major scalability bottlenecks of explicit segmentation or event detection. The central insight is topological: hybrid systems’ discontinuities arising from mode switching can be “glued” to form a piecewise smooth quotient manifold, on which the overall flow becomes spatially continuous. CHyLL operationalizes this known theoretical result, by: (1) learning a singularity-free neural embedding of the quotient manifold in a higher-dimensional latent space, (2) concurrently learning the continuous flow within this embedded space, and (3) enabling prediction and control of hybrid dynamics without explicitly modeling discrete modes or resets. Experiments show that CHyLL can reconstruct hybrid system flows with high accuracy, recover topological invariants, and be applied to stochastic optimal control problems.

**Strengths:**

(1) I find paper generally well written, besides few minor issues listed bellow. I like the clarity of motivation, transparency of novelties and  appreciate the balance in presenting both intuition and technical complexity.

(2) Up to my knowledge, CHyLL appears genuinely novel in its topological formulation of hybrid system learning. The reinterpretation of mode switches as gluing operations forming a quotient manifold and the learned embedding that makes hybrid flows continuous is non-standard path. While some ideas overlap with Neural ODE extensions and manifold-learning methods, no prior work explicitly connects hybrid system topology, latent manifold embedding, and continuous neural flow learning in a single approach.

(3) I believe that making the embedding theorem (Simic et al. 2005) from differential topology operational within Neural ODEs can inspire new methods that up to known were not able to successfully tackle hybrid systems.

(4) Experimental setup is generally appropriate.

**Weaknesses:**

(1) Presentation:
- Section 3: I find that introduction of main concepts like guards and resets should be smoother. Before jumping to formal Definitions, authors can use Figure 2 to introduce these concept first informally to build the intuition. e.g. in simple terms, what is the role of $q$.
- Section 2: I feel that related work section would be easier to parse after the intuition and notation on hybrid systems is current Section 3.
- Levenberg-Marquardt: Due to unclear Experimental conclusions (lack of std),  this aspect can be either avoided or emphasised. Computational overhead of using it should be discussed.

(2) Experiments: Standard deviations across trials is missing in experiments of Section 6.2. This makes it hard to conclude which version of the proposed method (w/o LM) is better performing, and diminishes the impact of the conclusions.

(3) Minor:
- should read "data" in line 182
- "be" lacking in line 189
-  notation should be ${\cal L}_c(\theta)$ in line 299

While my current score reflects the above weaknesses, I am happy to revise it if the rebuttal is successful.

**Questions:**

Given the good performance of DynamicAE of Lusch et al. 2018 in experiment of Section 6.3, what do authors think of adding additional  discussion on combining CHyLL with Koopman-based method. Namely, instead of using parametrising the vector filed and using Eq. (5) to evolve the latent dynamics, Koopman approach would learn linear evolution. Also, since DynamicAE is known to fail in modelling evolution in longer time-horizon, one can think of combining the CHyLL with representation learning for Koopman/Transfer operators, e.g. of references bellow, to learn appropriate representations of hybrid systems.

Han et al. Deep learning of Koopman representation for control, IEEE CDC2020
Kostic et al. Learning invariant representations of time-homogeneous stochastic dynamical systems, ICLR2024
Kostic et al. Neural conditional probability for uncertainty quantification, NeurIPS2024
Jeong et al. Efficient Parametric SVD of Koopman Operator for Stochastic Dynamical Systems, arXiv preprint 2025

---

> ### Author Response · Authors · 2025-11-24
> **Responses (Part 1)**
>
> We would like to thank the reviewer for the appreciation of the novelty of this work. We also thank the reviewer for the inspiring ideas about the Koopman operator.
>
> We here include:
> 1. Required reapted experiments.
> 2. New experiment on a robotic system in the Table. 1 and Fig. 7.
> 3. Ablation studies in Sec. 6.2 and Table. 2.
> 4. Responses to each question
>
> For the weakness:
>
> **Presentations:**
> 1. **Section 3 about the smoothness of introducing hybrid systems:** We have moved Fig. 2 to Sec. 2 for an intuition when introducing the hybrid system literature. We also added more captions to elaborate on the components of the hybrid automaton.
> 2. **Section 2 about clarity:** The aforementioned change is also intended to fix this problem.
> 3. **Levenberg-Marquardt:** The main reason to use LM is that the whole pipeline can be decoder-free as the encoder $M\rightarrow \mathbb{R}^{2n}$ is an almost injective function. Thus, solving the LM is a way to compute the inverse implicitly. Ideally, solving this LM can have a better result provided that the initial guess is sufficiently good. However, we noted that several factors will make the performance degenerate: 1) the latent integrator is not intrinsic, which means the rollout, i.e., $\hat{z}$, may not be on the manifold, i.e., $\mathrm{Img}E(\cdot)$. 2): as the LM is a projection problem on a nonconvex set (no general solution to this problem with guaranteed performance), the curvature of the latent / how the manifold is glued will result in a totally different optimization landscape or trajectories.
>
> We find that when the latent flow prediction is not sufficiently good, the performance on LM will degrade. The conformal loss is also crucial to improving the LM/decoder performance as it significantly reduces the latent manifold distortion.
>
> We use the analytical solution for the Jacobian of MLP, the computational burden is 10x more lower than autodiff as we only compute Jacobian in the forward inference of MLP.
>
>
> **Experiment:**
>
> 1. We have added the repeated test for evaluations of encoder/LM in 5 random experiments with the same parameters in **Table. 1**. We report the RMSE (std.dev of RMSE) across the five trials:
>
> **Bouncing Ball:**
> Decoder: 0.271(0.072) | LM: 0.237(0.0629)
>
> **Torus:**
> Decoder: 0.0165(0.00927) | LM: 0.0164(0.00989)
>
> **Klein:**
> Decoder: 0.0237(0.00567) | LM: 0.0228(0.00586)
>
> We show that in the Boucing Ball and Klein Bottle, the LM significantly improves the performance, while in the Torus case, LM provides a slightly better mean RMSE with slightly bigger variance. As mentioned before, improving the LM result requires 1) a high-quality latent rollout and 2) a well-conditioned encoder.
>
> Other baselines in these examples have obvious failures (divergence, failure to capture the pattern, or penetration).
>
> 2. We have added one more example on a 3-link walker with higher higher-dimensional state space (6 DOF), nonlinear dynamics, and multi-modes (one leg stance, two leg stance, flying, slip...). The proposed method outperforms the existing method.
>
> For methods without a failed pattern, we tested them on five different trials and reported the MSE(std.dev of MSE) in Table 1. The revised manuscript:
>
> i. CHyLL:** Decoder: 0.233(0.0134) | LM: 0.243(0.0137)
>
> ii. Neural ODE: 0.275(0.0610)
>
> iii. RNN-ODE: 0.253 (0.0770)
>
> iv Koopman (Diverge): 151.9
>
> v. AutoEncoder (Diverge): 0.567
>
> vi. Event ODE (Ill-conditioned)
>
> We can see that the proposed CHyLL outperforms the other methods in accuracy and in capturing the correct pattern.
>
> We note that this scenario is more nonlinear that resulting in worse $\hat{z}$ prediction than the previous examples. The initial guess of LM is degenerate, thus resulting in worse refinement solutions. **The result is presented in a new line in the Table. 1 and Figure. 7.**
>
> **Minor:**
> We have corrected these typos and marked them blue in the paper.

---

> ### Author Response · Authors · 2025-11-24
> **Responses (part 2)**
>
> **Questions about Koopman operator:**
> We thank the reviewer for pointing out the related work and for the insightful question. We have included this literature and added more discussions.
>
> Combining the two methods is an interesting extension for future work. Our perspective is that there are two possible ways.
>
> 1. **Spectral theorem–based approach:**
>    We can use a high-dimensional linear space as the latent embedding. This allows us to retain the convergence guarantees associated with infinite-dimensional Koopman operators. Then we could (1) add the gluing loss when training the encoder, and (2) decouple the training of the encoder and decoder. In this way, we can ensure that the latent space does not exhibit discontinuities, unlike conventional Koopman operators, which do not enforce continuity in the latent representation.
>
> 2. **Embedding theorem–based approach:**
>    Here, CHyLL can be used with a much higher latent dimension to encourage an approximately isometric embedding. According to the Nash embedding theorem, an **isometric** and **smooth** embedding exists for any $n$-dimensional compact manifold. The required latent dimension is $m = \frac{n(3n + 11)}{2}$ for compact manifold or $m = \frac{n(n+1)(3n + 11)}{2}$ for non-compact manifold. To ensure that $E(\cdot): M \rightarrow \mathbb{R}^m$ is isometric. In this case, the latent space can theoretically preserve the curvature of the original manifold \(M\) while avoiding singularities. In typical scenarios where the data evolves on Euclidean space (with the identity metric), this yields a latent space with **no** local distortion. Although the bound on $m$ is quite high, this intuition aligns with Koopman theory: higher-dimensional embeddings help reduce nonlinearity and make dynamics learning easier. **This idea is tested in the revision with larger latent dimension with improvement in the performance. See Table. 2.**
>
> In summary, combining Koopman theory with embedding-theorem-based–based ideas offers a promising direction for obtaining more structured and well-behaved latent representations for hybrid systems. The key principles are (1) enforcing continuity guaranteed by the underlying topology, and (2) using higher-dimensional latent spaces to reduce nonlinearity.

---

> > ### Comment · Reviewer_488A · 2025-11-25
> >
> > I thank the authors for their replies that addressed all my comments. In particular, I appreciate the additional discussion related to Koopman operator framework. I find that the revised paper has improved, and, accordingly,  I adapt my score.

---

### Author Response · Authors · 2025-12-01
**Author Final Remark**

We sincerely thank the area chair and all reviewers for their thoughtful feedback and constructive suggestions.

We **especially thank the area chair** for the additional effort and time at this special moment!

We are encouraged that **two reviewers explicitly commented that they will raise their scores** as the responses addressed all their concerns. The remaining reviewer **maintained a positive stance**. This convergence of opinions reflects a consensus on the contribution and potential impact of our work.

Our paper proposed the **CHyLL** (**C**ontinuous **Hy**brid System **L**earning in **L**atent Space) framework that learns the *discontinuous* flow of hybrid systems (governed by continuous vector fields and discrete state reset) as *continuous* latent flows. CHyLL designs the loss function based on rigorous topology and geometry theory to guarantee the latent flow is continuous and admits a singularity-free representation. Using CHyLL, we can avoid fundamental limits in existing methods, such as the assigning trajectory segments to different modes (exponentially hard) and the ill-conditioning in event-function-based simulations. CHyLL only requires time-series data without labeling or discrete actions.

The reviewer **488A** maintained an **initial positive score**, highlighted that this work is “**genuinely novel**” and **commented** that "*I thank the authors for their replies that addressed all my comments. In particular, I appreciate the additional discussion related to the Koopman operator framework. **I find that the revised paper has improved, and, accordingly, I adapt my score**.*"

The reviewer **wC3u** also **commented** that "*The authors have addressed my main concerns, I have **raised my score** accordingly.*" We have point-to-point responses to all the questions and provided the required ablation study, new experiment, and more discussions. **wC3u** increased the score to **positive** after our responses.

The reviewer **EUQm** maintained an **initial positive score**, highlighted that CHyLL as “**drawing on profound mathematical theories to guide the methodology design**” and described this work as “**a groundbreaking method**”. The main concern of the EUQm is about several mathematical definitions and requests for more experiments. We have added these clarifications in the footnotes and one additional experiment in the revised paper. We expect EUQm to raise the score or maintain the current positive opinion.

With the clarifications and improvements made during the rebuttal, we hope the area chair will find the manuscript suitable for inclusion in the ICLR program. It would be a great honor for us to contribute to this vibrant and forward-looking community.

---

### Note · Authors · 2026-01-05

I have read and agree with the venue's withdrawal policy on behalf of myself and my co-authors.